# Simulation Study on Heat Generation Characteristics of Lithium-Ion Battery Aging Process

**Rui Huang** [1,2,3]**, Yidan Xu** [4]**, Qichao Wu** [4]**, Junxuan Chen** [1,2]**, Fenfang Chen** [4] **and Xiaoli Yu** [3,4,*]

1    Key Laboratory of Clean Energy and Carbon Neutrality of Zhejiang Province, Hangzhou 310063, China
2    Jiaxing Research Institute, Zhejiang University, Jiaxing 314031, China
3    Key Laboratory of Smart Thermal Management Science & Technology for Vehicles of Zhejiang Province, Taizhou 317200, China
4    College of Energy Engineering, Zhejiang University, Hangzhou 310027, China
*    Correspondence: yuxl@zju.edu.cn

**Abstract:** Lithium-ion battery heat generation characteristics during aging are crucial for the creation of thermal management solutions. The heat generation characteristics of 21700 (NCA) cylindrical lithium-ion batteries during aging were investigated using the mathematical model that was created in this study to couple electrochemical mechanisms, heat transfer, and aging loss. These findings indicate that, at the same operating current, the heat generation power of the cell increased significantly with battery aging. This increase was primarily due to the energy loss caused by the growth of the solid–electrolyte interface (SEI) and a reduction in the negative porosity and other physical characteristics of the SEI, such as its ionic conductivity and molar volume, which also had an impact on the heat generation power. By investigating the variations in battery heat generation in different aging modes, the electrochemical mechanisms underlying the effects of aging on battery heat generation can be comprehended in depth.

**Keywords:** lithium-ion battery; electrochemical–aging–thermal model; heat generation characteristics; SEI; aging modes

## 1. Introduction

Due to the rising energy density of lithium-ion batteries, their performance, lifetime, and thermal stability are substantially influenced by temperature, and thermal management has become a pressing concern [1–4]. The temperature of a lithium-ion battery is dictated by its internal heat generation power and external heat transfer power; therefore, knowing the heat generation characteristics of lithium-ion batteries is essential for effective and precise thermal management [5,6]. The internal heat generation characteristics of the cell can be studied thoroughly and efficiently using a numerical model based on the cell's mechanism and a vast amount of experimental data.

According to published research, the heat generation characteristics of lithium-ion batteries are complex and influenced by many factors [7,8]. Bernardi et al. [9] carried out an analysis of the thermodynamic energy balance inside the battery and pointed out that the heat generated during the use of the battery mainly comes from Joule heat, reaction heat, mixing heat, and phase change heat. On this basis, Bahiraei et al. [10] and Chiew et al. [11] coupled the battery heat generation model with a pseudo-two-dimensional electrochemical model for thermal management system scheme design. Lithium-ion batteries were subject to significant aging during operation, and the aging of the battery had an important impact on the parameters that relate to the internal chemistry of the lithium-ion battery. Yi et al. [12,13] developed a two-dimensional distribution model of potential, current density, and heat generation for pouch cells to enable the computation of the battery's age-related heat generation. As a major side reaction leading to cell aging, the SEI growth reaction has an important influence on the heat generation of the cell during aging by

its products [14–16]. Darcovich et al. [17] combined the finite volume model and single particle model to establish the two-dimensional model of a prismatic cell and coupled it with a simplified SEI growth model to study changes in the heat of polarization, ohmic heat, and electrochemical reaction heat of the cell during aging. Tang et al. [18] established an electrochemical–thermal coupling model for pouch cells, considering the electrolyte reduction decomposition reaction and the growth of the SEI and analyzed the changes in reversible heat, polarization heat, ohmic heat, and the total heat generation during the charging and discharging of batteries in different aging states; they found that the heat generation power increased during the charging and discharging of aged batteries, but that the total heat generation decreased. Researchers have undertaken experimental investigations on variations in heat generation during battery aging, but none of them has focused on the impacts of the battery aging process on heat generation, nor have they studied the effects of aging on the characteristics of heat generation.

In this paper, an electrochemical–aging–thermal coupling model of a lithium-ion battery was proposed. Model parameters that are sensitive to temperature change were estimated by comparing modeling findings with experimental data. After detailed model validation at varied operating temperatures and charge–discharge rates, the changes in the heat-generating components over the aging process and the influences of SEI physical characteristics on the heat-generation process were simulated and reviewed. The mechanism of the effect of different aging processes on heat generation was investigated.

## 2. Experiments

A commercial 21700 NCA battery ($\Phi21$ mm $\times$ 70 mm cylinder battery, capacity 4.9 Ah, Si-C anode material, Li ($Ni_{0.8}Co_{0.15}Al_{0.05}$) $O_2$ cathode material) was experimentally characterized. A NEWARE (CT4008-5V12A; Shenzhen, China) test system, 8-channel A-to-D converter, and a computer data logger were used to monitor the charge–discharge current and battery voltage. A real-time graph was created and stored on a computer. The battery was placed in a temperature control box for operating at different temperatures.

The complete set of temperature measurement equipment consisted of a T-type thermocouple, an NI 9214 (NI, Austin, TX, USA) thermocouple board, an NI cRIO 9037 (NI, Austin, TX, USA) data acquisition controller, etc. The T-type thermocouple probe was stuck on the central site of the batteries, and the batteries were continuously charged and discharged with a certain constant current. Lastly, the surface temperature change during the operation was monitored. During the full process, the battery surface temperature was measured and recorded in situ by a multi-channel temperature logger.

Pulse tests at different ambient temperatures (23 °C, 40 °C) and different currents (0.5 C, 1 C, 2 C) were performed to characterize the dynamic performance of the lithium-ion battery. It should be noted that the temperature-dependent parameters, such as lithium-ion diffusivity in solid and solution phases, and the reaction rate could be extracted from these experimental data.

## 3. Model Development

### 3.1. Model Assumptions and Calculation Domain

This study developed an electrochemical–aging–thermal model for cylindrical 21700 NCA lithium-ion batteries based on the pseudo-two-dimensional model of Doyle et al. [19–21]. The principal model assumptions are displayed as follows:

(1) The intricate winding structure within the cell was disregarded and the cell was treated as a uniform cylinder.
(2) Current collector effects on lithium-ion transport and heat transfer were disregarded.
(3) The change in the entropic coefficient throughout the battery aging process was not taken into account.
(4) Adverse effects during the operation were disregarded.

Figure 1 shows a sketch of the winding structure of the cylindrical cell [22] and the schematic calculation domain of the one-dimensional (1D) battery model. In Figure 1, $L_{Cu}$,

$L_n$, $L_{sep}$, $L_p$, and $L_{Al}$ denote the thickness of the negative collector (copper foil), negative electrode, separator, positive electrode, and positive collector (aluminum foil), respectively.

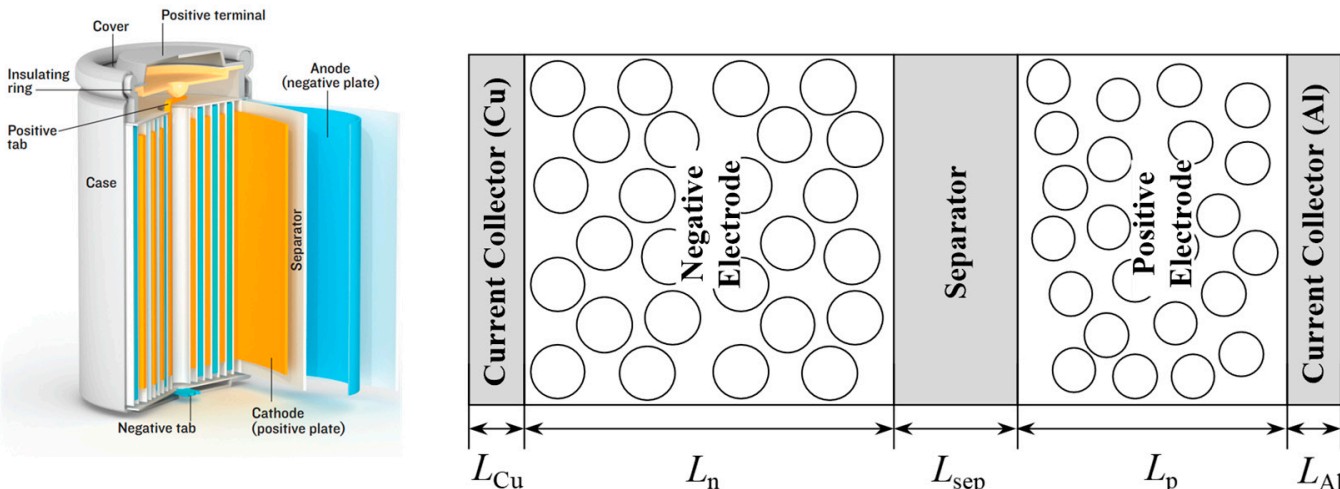

**Figure 1.** Intrinsic cell structure [22] and model calculation domain.

### 3.2. Electrochemical Model

#### 3.2.1. Lithium-Ion Concentration Distribution

Assuming that the positive and negative active material particles are spherical and that the lithium ions only have a concentration gradient in the radial direction of the spherical particles, the distribution of lithium ions in the solid phase radial direction according to Fick's second law is given by the equation:

$$\frac{\partial c_s}{\partial t} = \frac{D_{s,i}}{r^2}\frac{\partial}{\partial r}\left(r^2\frac{\partial c_s}{\partial r}\right) \tag{1}$$

where $c_s$ is the lithium-ion concentration in the solid phase, $D_{s,i}$ is the solid phase diffusivity, and $r$ is the radius distance variable of the particle.

The following equation describes the concentration change in lithium ions in the electrolyte (solution phase) due to the combined effects of diffusion and electromigration:

$$\frac{\partial}{\partial t}\varepsilon_{e,i}c_e = \frac{\partial}{\partial x}\left(D_{e,i}^{eff}\frac{\partial c_e}{\partial x}\right) + \frac{(1-t^+)j_{tot}}{F} \tag{2}$$

where $\varepsilon_{e,i}$ is the solution phase volume fraction, $c_e$ is the lithium-ion concentration in the solution phase, $x$ is the distance variable through a cell component, $t^+$ is the transference number of Li-ion species dissolved in a solution, $j_{tot}$ is the total current density, and $F$ is Faraday's constant.

The relationship between the effective diffusivity $D_{e,i}^{eff}$ and the diffusivity $D_{e,i}$ of the solution phase is

$$D_{e,i}^{eff} = D_{e,i}\varepsilon_e^{\beta_i} \tag{3}$$

#### 3.2.2. Electric Potential Distribution

Based on Ohm's law and the distribution of potential in solid and solution phases along the thickness direction of the electrode, the equation of solid phase potential $\Phi_s$ distribution can be derived.

$$\frac{\partial}{\partial x}\left(\sigma_i^{eff}\frac{\partial}{\partial x}\phi_s\right) = j_{tot} \tag{4}$$

The relationship between the effective conductivity $\sigma_i^{eff}$ and the conductivity $\sigma_i$ of the solid phase is

$$\sigma_i^{eff} = \sigma_i\varepsilon_{s,i} \tag{5}$$

The potential distribution equation for the solution phase is derived using the law of conservation of charge and Ohm's law:

$$\frac{\partial}{\partial x}\left(\kappa^{\text{eff}}\frac{\partial}{\partial x}\phi_e\right) = -\frac{\partial}{\partial x}\left(\kappa_D^{\text{eff}}\frac{\partial}{\partial x}\ln c_e\right) - j_{\text{tot}} \tag{6}$$

$$\kappa_D^{\text{eff}} = \frac{2RT\kappa^{\text{eff}}}{F}\left(1 - t^+\right) \tag{7}$$

$$\kappa^{\text{eff}} = \kappa\varepsilon_{e,i}^{\beta_i} \tag{8}$$

where $\Phi_e$ is the solution phase potential, $\kappa^{\text{eff}}$ is the effective ionic conductivity of the electrolyte, $\kappa_D^{\text{eff}}$ is the effective ionic diffusion conductivity, $R$ is the gas constant, and $T$ is the cell temperature.

### 3.2.3. Electrode Reaction Kinetics

The Butler–Volmer kinetic equation can be used to describe the current density $j_{\text{int}}$ of the lithium embedding and de-embedding reaction:

$$j_{\text{int}} = a_{s,i}i_{0,i}\left\{\exp\left(\frac{\alpha_{p,\text{int}}F}{RT}\eta_{\text{act,int}}\right) - \exp\left(-\frac{\alpha_{n,\text{int}}F}{RT}\eta_{\text{act,int}}\right)\right\} \tag{9}$$

$$\eta_{\text{act,int}} = \phi_s - \phi_e - \frac{j_{\text{tot}}}{a_{s,i}}R_{\text{film}} - U_{\text{eq},i}(\theta_i) \tag{10}$$

$$i_{0,i} = Fk_{0,i}c_e^{\alpha_P}\left(c_{s,i}^{\max} - c_{s,i}^{\text{surf}}\right)^{\alpha_P}\left(c_{s,i}^{\text{surf}}\right)^{\alpha_n} \tag{11}$$

where $a_{s,i}$ is the specific surface area, $i_{0,i}$ is the exchange current density, $\alpha_p$, and $\alpha_n$ are the transfer coefficients for the positive and negative current, $\eta_{\text{act,int}}$ is the activation overpotential of the reaction, $R_{\text{film}}$ is the film resistance on the surface of the active material particles, $U_{\text{eq},i}$ is the open-circuit voltage, and $k_{0,i}$ is the reaction rate constant.

The positive and negative open-circuit voltages are related to their local charge states $\theta_i$, defined as the ratio of the lithium-ion concentration on the surface of the active material $c_{s,i}^{\text{surf}}$ to its maximum lithium ion concentration $c_{s,i}^{\max}$.

$$\theta_i = \frac{c_{s,i}^{\text{surf}}}{c_{s,i}^{\max}} \tag{12}$$

### 3.3. Aging Model

#### 3.3.1. Loss of Lithium-Ion Inventory (LLI)

During the operation, solvent molecules in the electrolyte, such as ethylene carbonate (EC), conducted a redox reaction on the surface of the negative electrode's active material to generate an SEI. Based on previous studies [23–26], this study assumed that the result of the SEI growth reaction in a battery was $Li_2CO_3$. In addition to the reaction kinetics, the diffusion of solvent molecules within the existing SEI limited the current density of the SEI growth reaction. Safari et al. [27] established a model for the SEI growth reaction, including diffusion constraints based on the Tafer formulation of the aforementioned process.

$$j_{\text{SEI}} = -Fk_{0,\text{SEI}}c_{\text{EC}}^s\exp\left[-\frac{\alpha_{n,\text{SEI}}F}{RT}\eta_{\text{act,SEI}}\right] \tag{13}$$

$$\eta_{\text{act,SEI}} = \phi_s - \phi_e - U_{\text{eq,SEI}} - \frac{R_{\text{film}}j_{\text{tot}}}{a_{s,n}} \tag{14}$$

The diffusion process of the electrolyte determines the link between the $c_{\text{EC}}^{\text{s}}$ and EC concentration:

$$-D_{\text{EC}}\frac{c_{\text{EC}}^{\text{s}} - c_{\text{EC}}^{0}}{\delta_{\text{film}}} = -\frac{j_{\text{SEI}}}{F} \tag{15}$$

where $c_{\text{EC}}^{\text{s}}$ is the EC concentration diffused to the surface of the negative active particle, $c_{\text{EC}}^{0}$ is the EC concentration in the electrolyte, $\delta_{\text{film}}$ is the thickness of the film on the surface of the active particle, and $D_{\text{EC}}$ is the diffusion coefficient of EC in the film.

Combining Equations (13) and (15) produces

$$j_{\text{SEI}} = -\frac{c_{\text{EC}}^{0}F}{\dfrac{\delta_{\text{film}}}{D_{\text{EC}}} + \dfrac{1}{k_{0,\text{SEI}}\exp\left[-\dfrac{\alpha_{\text{n,SEI}}F}{RT}\eta_{\text{act,SEI}}\right]}} \tag{16}$$

The total reaction current density on the surface of the active material particles is equal to the sum of the current densities of the SEI growth reaction and the de-embedded lithium reaction.

$$j_{\text{tot}} = j_{\text{int}} + j_{\text{SEI}} \tag{17}$$

According to the principle of mass conservation, the SEI's growth rate is

$$\frac{\partial c_{\text{SEI}}}{\partial t} = -\frac{j_{\text{SEI}}}{2F} \tag{18}$$

The SEI film growth response causes the thickening of the active material's surface film. According to the conservation of mass, and assuming that the negative active material particles are spherical and the surface film is uniformly covered, the rate of change in the active material surface film thickness $\delta_{\text{film}}$ is as follows:

$$\frac{\partial \delta_{\text{film}}}{\partial t} = \frac{\partial c_{\text{SEI}}}{\partial t} \cdot \frac{M_{\text{SEI}}}{a_{\text{s,n}}\rho_{\text{SEI}}} = -\frac{j_{\text{SEI}}M_{\text{SEI}}}{2Fa_{\text{s,n}}\rho_{\text{SEI}}} \tag{19}$$

$$\frac{\partial R_{\text{film}}}{\partial t} = -\frac{1}{\kappa_{\text{SEI}}} \cdot \frac{j_{\text{SEI}}M_{\text{SEI}}}{2Fa_{\text{s,n}}\rho_{\text{SEI}}} \tag{20}$$

where $M_{\text{SEI}}$ is the molar mass and $\rho_{\text{SEI}}$ is the density of the SEI.

SEI growth leads to a decrease in the porosity of the negative electrode, as described by the following equation:

$$\frac{\partial \varepsilon_{\text{s,n}}}{\partial t} = -a_{\text{s,n}}\frac{\partial \delta_{\text{film}}}{\partial t} \tag{21}$$

3.3.2. Loss of Active Material (LAM)

By analyzing the differential voltage of the battery aging process, a high linearity was found for the active material loss indicator $G_{\text{DVA\_LAM}}$ and the cumulative battery power throughput $\psi$ after taking the logarithm, respectively. Therefore, after a linear fit, it can be converted into a power function to represent the relationship between the active material loss index and the cumulative battery power throughput, and the rate of change in the solid phase volume fraction as the active material loss rate can be expressed using the following equation:

$$\Delta\varepsilon_{\text{s},i} = \mu A \cdot \psi^{B} \tag{22}$$

where $\psi$ is the cumulative battery power throughput, and both $A$ and $B$ are constants.

*3.4. Thermal Model*

The reversible reaction heat, heat of polarization, ohmic heat, heat of SEI growth, and contact resistance heat are among the heat-generating factors that were taken into account

by the model [28]. When the aforementioned five heating elements were combined, the battery's overall heat generation power was

$$\dot{q}_{\text{tot}}(x,t) = \dot{q}_{\text{rea}} + \dot{q}_{\text{act}} + \dot{q}_{\text{ohm}} + \dot{q}_{\text{film}} + \dot{q}_{\text{c}} \tag{23}$$

The cylindrical coordinate system can be employed in the single-cell model since the object of this investigation is a cylinder. The internal heat transfer equation for the cell can be obtained by simplifying it to a cylinder with a homogeneous interior material and equally distributed heat production:

$$\rho_{\text{batt}}C_{\text{p,batt}}\frac{\partial T}{\partial t} = \frac{1}{r}\frac{\partial}{\partial r}\left(\lambda_{\text{batt,r}}r\frac{\partial T}{\partial r}\right) + \frac{1}{r^2}\frac{\partial}{\partial \varphi}\left(\lambda_{\text{batt},\varphi}\frac{\partial T}{\partial \varphi}\right) + \frac{\partial}{\partial z}\left(\lambda_{\text{batt,z}}\frac{\partial T}{\partial z}\right) + \dot{q}_{\text{batt}} \tag{24}$$

$$\dot{q}_{\text{batt}} = \frac{\int_0^{L_{\text{batt}}} \dot{q}_{\text{tot}}A_c\mathrm{d}x}{\pi R_{\text{batt}}^2 l_{\text{batt}}} \tag{25}$$

where $\rho_{\text{batt}}$ is the density of the cell, $C_{\text{p,batt}}$ is the heat capacity of the cell, $\lambda_{\text{batt,r}}$, $\lambda_{\text{batt},\varphi}$, and $\lambda_{\text{batt,z}}$ are the radial, circumferential, and axial thermal conductivities of the cell, respectively, $\dot{q}_{\text{batt}}$ is the average power of heat generation per unit volume of the cell, $L_{\text{batt}}$ is the total electrode thickness (including the collector), $R_{\text{batt}}$ is the outer diameter of the cell, $l_{\text{batt}}$ is the height of the cell, and $A_c$ is the area of the positive (both sides) that has an opposing negative.

The power of the external surface's heat transmission can be defined as follows when the external medium is air, coolant, and other fluids, with the core and fluid convective heat transfer:

$$-\lambda_{\text{batt,r}}\frac{\partial T}{\partial r}\bigg|_{r=R_{\text{batt}}} = h_{\text{cov}}[T(R_{\text{batt}},t) - T_{\text{amb}}] \tag{26}$$

where $h_{\text{cov}}$ is the heat transfer coefficient and $T_{\text{amb}}$ is the ambient temperature.

### 3.5. Solution Method

The coupling relationships between the above electrochemical, aging, and thermal models are shown in Figure 2. The model calculations were performed using the finite element commercial software COMSOL MULTIPHYSICS 6.0.

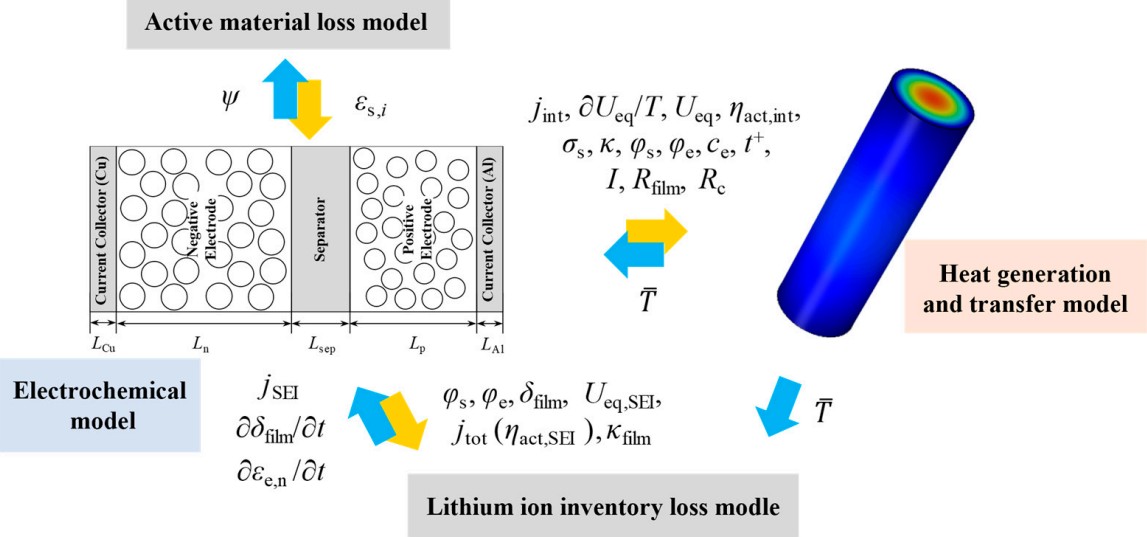

**Figure 2.** Coupling relationship between submodels.

## 4. Model Parameterization and Validation

### 4.1. Parameterization

4.1.1. Basic Design Parameters

For simulation research, the physical parameters of the simulation object are crucial. The parameters of the model were determined using experimental measurements, computations, and estimates. The electrochemical–aging–thermal design parameters for the 4.9 Ah NCA battery are provided in Tables 1–3.

**Table 1.** Parameters used in the electrochemical model.

| Parameters | Negative Electrode | Separator | Positive Electrode |
|---|---|---|---|
| $L_i/\text{m}$ | $5 \times 10^{-6}$ | - | $5 \times 10^{-6}$ |
| $\sigma_{s,i}/\text{S·m}^{-1}$ | $6.3 \times 10^{7}$ [6] | - | $3.8 \times 10^{7}$ [6] |
| $L_i/\text{m}$ | $88 \times 10^{-6}$ | $8 \times 10^{-6}$ | $62 \times 10^{-6}$ |
| $R_i/\text{m}$ | $9 \times 10^{-6}$ | - | $5 \times 10^{-6}$ |
| $\varepsilon_{s,i}$ | 0.71 | - | 0.703 |
| $\varepsilon_{e,i}$ | 0.21 | 0.45 | 0.19 |
| $c_{e,0}/\text{mol·m}^{-3}$ | 1200 [29] | 1200 [29] | 1200 [29] |
| $c_{s,i}^{\max}/\text{mol·m}^{-3}$ | 34,507 [29] | - | 49,000 [29] |
| $\alpha_{a,\text{int}}, \alpha_{c,\text{int}}$ | 0.5 [6] | - | 0.5 [6] |
| $\beta$ | 1.5 | 1.5 | 1.7 |
| $D_{s,i}/\text{m}^2\text{·s}^{-1}$ | $9 \times 10^{-14}$ [30] | - | $1.5 \times 10^{-14}$ [30] |
| $\sigma_{s,i}/\text{S·m}^{-1}$ | 100 [29] | - | 3.8 [29] |
| $i_{0,\text{ref}}/\text{A·m}^2$ | 0.75 | - | 2 |
| $E_{aD,i}/\text{J·mol}^{-1}$ | 48,000 | - | 22,000 |
| $E_{ak,i}/\text{J·mol}^{-1}$ | 36,000 | - | 30,000 |
| $t_+$ | | 0.363 [6] | |
| $T_{\text{ref}}/\text{K}$ | | 298.15 | |
| $R_{\text{film},0}/\Omega\text{·m}^2$ | | 0.001 [28] | |
| $A_c/\text{m}^2$ | | 0.12 | |
| $F/\text{C.mol}^{-1}$ | | 96,487 | |
| $R/\text{J.(mol·K)}^{-1}$ | | 8.314 | |

**Table 2.** Parameters used in the aging model.

| Parameters | Value |
|---|---|
| $c_{\text{EC}}^{0}/\text{mol·m}^{-3}$ | 4500 [31] |
| $U_{\text{eq, SEI}}/\text{V}$ | 0.4 [28] |
| $\alpha_{c,\text{ SEI}}$ | 0.5 [28] |
| $E_{a,Dsei}/\text{J·mol}^{-1}$ | 30,000 |
| $E_{a,ksei}/\text{J·mol}^{-1}$ | 35,000 |
| $\sigma_{\text{film}}/\text{S·m}^{-1}$ | $5 \times 10^{-6}$ [31] |
| $M_{\text{SEI}}/\text{kg·mol}^{-1}$ | 0.07388 |
| $\rho_{\text{SEI}}/\text{kg·m}^{-3}$ | 2110 [25] |

**Table 3.** Parameters used in the thermal model.

| Parameters | Value |
|---|---|
| $C_{p,\text{batt}}/\text{J·kg}^{-1}\text{·K}^{-1}$ | 880 |
| $\lambda_{\text{batt},r}/\text{W·m}^{-1}\text{·K}^{-1}$ | 0.7 |
| $\lambda_{\text{batt},\varphi}/\text{W·m}^{-1}\text{·K}^{-1}$ | 10.5 |
| $\lambda_{\text{batt},z}/\text{W·m}^{-1}\text{·K}^{-1}$ | 10.5 |
| $\rho_{\text{batt}}/\text{kg·m}^{-3}$ | 2846 |
| $h_{\text{cov}}/\text{W·m}^{-2}\text{·K}^{-1}$ | 18 |
| $l_{\text{batt}}/\text{m}$ | 0.0700 |
| $R_{\text{batt}}/\text{m}$ | 0.0105 |

### 4.1.2. Kinetic Parameters

In this paper, the electrochemical–aging–thermal model was utilized to investigate the features of lithium-ion battery heat generation. As the diffusion coefficients of lithium in the positive and negative active materials and the reaction rates of the positive and negative electrodes were strongly impacted by temperature, Arrhenius' formula was typically used to characterize their relationship with temperature.

$$X_i(T) = X_i^{\text{ref}} \exp\left[\frac{E_{\text{a}X,i}}{R}\left(\frac{1}{T_{\text{ref}}} - \frac{1}{T}\right)\right] \tag{27}$$

Using the relationship described in [32,33], the diffusion coefficient $D_{\text{e}}$ and conductivity $\kappa$ of lithium ions in the electrolyte could be calculated.

$$D_{\text{e}} = 1470 \cdot \exp(0.00133c_e) \cdot \exp\left[\frac{-1690}{T}\right] \cdot \exp\left[\frac{-0.563c_e}{T}\right] \cdot 10^{-10} \text{ m}^2/\text{s} \tag{28}$$

$$\kappa = 10^{-4}c_e(-10.5 + 0.074T - 6.96 \times 10^{-5}T^2 + 6.68 \times 10^{-4}c_e \\ -1.78 \times 10^{-5}c_e T + 2.8 \times 10^{-8}c_e T^2 + 4.94 \times 10^{-7}c_{\text{e}}^2 - 8.86 \times 10^{-10}c_{\text{e}}^2 T)^2 \tag{29}$$

### 4.1.3. Thermodynamic Parameters

The potential difference of the electrode material is not only determined by the amount of lithium incorporated in different cell materials but also by the temperature. The model uses the following equation to calculate the open-circuit voltage of the electrode at various temperatures.

$$U_{\text{eq},i} = U_{\text{eq},i}^{\text{ref}} + \frac{\partial U_{\text{eq},i}}{\partial T}(T - T_{\text{ref}}) \tag{30}$$

Figure 3 depicts the open-circuit voltages (OCV) and entropic coefficient of Si–C negative and NCA positive at different states of charge (SOC).

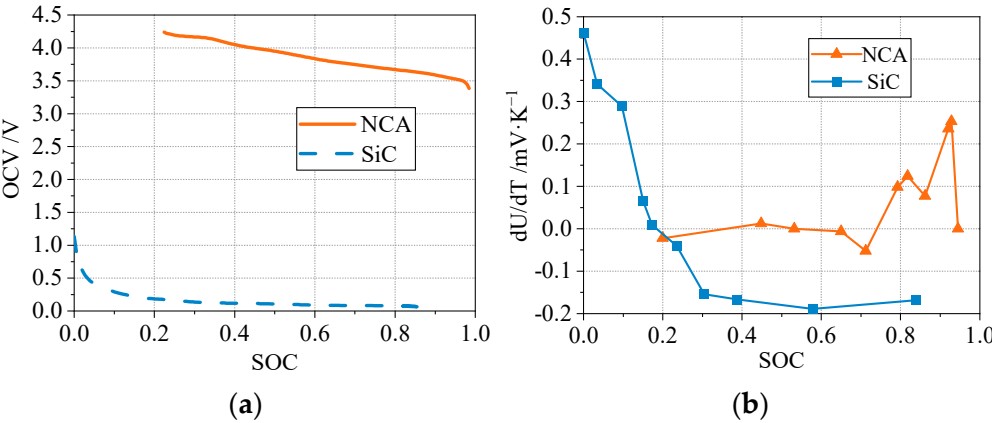

(**a**)   (**b**)

**Figure 3.** Thermodynamic parameters at different SOC: (**a**) open-circuit voltages and (**b**) the entropic coefficient.

### 4.1.4. Active Material Loss Model Calibration

According to Equation (22), and in conjunction with the experimental data, a functional link could be established between the loss index of positive and negative active materials and the accumulated battery power throughput under different cycle settings, as demonstrated by Equations (31)−(34). The loss of positive and negative active materials at

40 °C was less affected by the charging and discharging multiplier of the cycle; hence, the same relationship could be utilized for different multipliers.

$$\begin{cases} \Delta\varepsilon^{\mathrm{n}}_{s,i\_23C0.5D1} = 4.73 \times 10^{-5} \cdot \psi^{0.79} \\ \Delta\varepsilon^{\mathrm{p}}_{s,i\_23C0.5D1} = 6.93 \times 10^{-20} \cdot \psi^{4.58} \end{cases} \tag{31}$$

$$\begin{cases} \Delta\varepsilon^{\mathrm{n}}_{s,i\_23C1D1} = 8.15 \times 10^{-5} \cdot \psi^{0.90} \\ \Delta\varepsilon^{\mathrm{p}}_{s,i\_23C1D1} = 4.60 \times 10^{-16} \cdot \psi^{3.77} \end{cases} \tag{32}$$

$$\begin{cases} \Delta\varepsilon^{\mathrm{n}}_{s,i\_23C1D2} = 4.08 \times 10^{-4} \cdot \psi^{0.84} \\ \Delta\varepsilon^{\mathrm{p}}_{s,i\_23C1D2} = 8.62 \times 10^{-22} \cdot \psi^{5.37} \end{cases} \tag{33}$$

$$\begin{cases} \Delta\varepsilon^{\mathrm{n}}_{s,i\_40} = 7.5 \times 10^{-4} \cdot \psi^{0.55} \\ \Delta\varepsilon^{\mathrm{p}}_{s,i\_40} = 4.26 \times 10^{-8} \cdot \psi^{1.68} \end{cases} \tag{34}$$

### 4.2. Model Validation

To validate the accuracy of the electrochemical–aging–thermal coupling model produced previously, the calculated results of the model were compared to the experimental test results. Figure 4 compares the experimental and simulated voltage and surface temperature rise curves of the battery during constant current discharge under various operating conditions. Figure 5 compares the cell's heat production power under adiabatic conditions (initial temperature of 31 °C) with a continuous current discharge at a 0.5 C multiplier. The foregoing comparison with the experimental results demonstrates that the electrochemical–thermal coupling model proposed in this study could accurately simulate the electrochemical and heat production characteristics of the new cell under varying operating conditions.

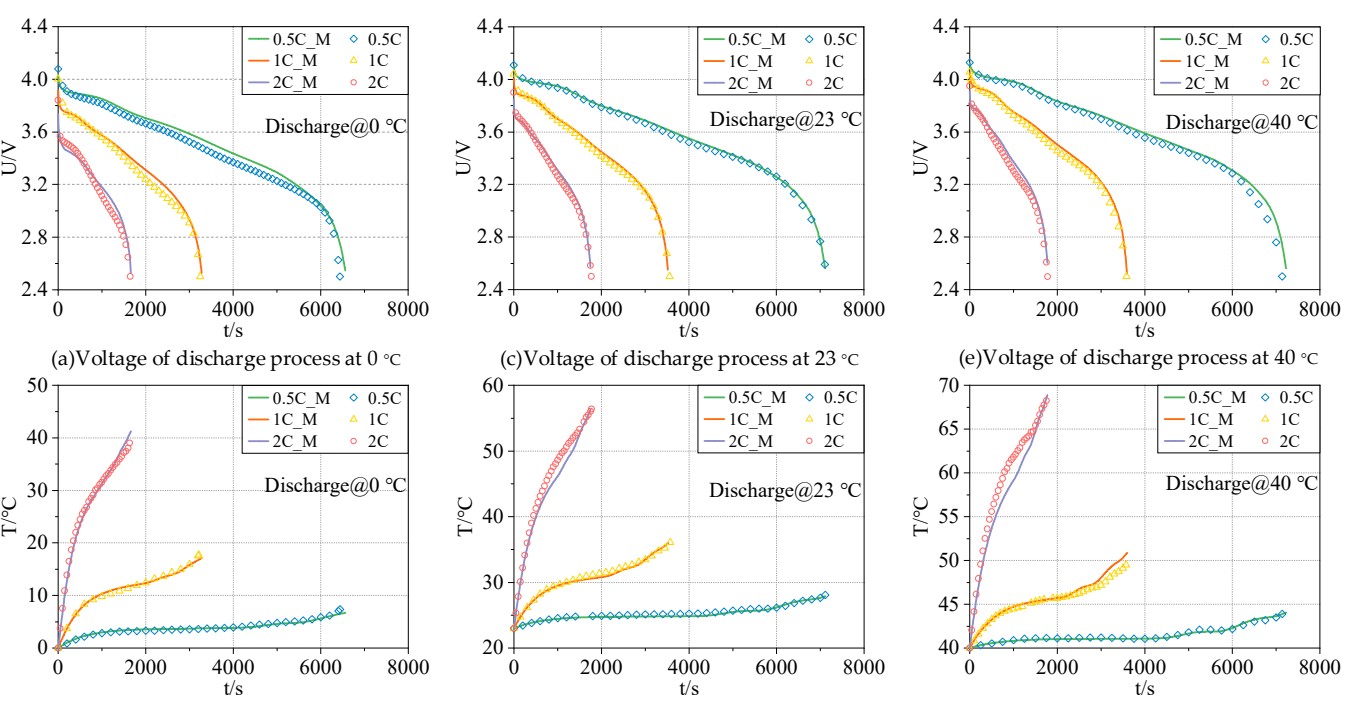

**Figure 4.** Validation of new battery discharge voltage and surface temperature rises under different conditions.

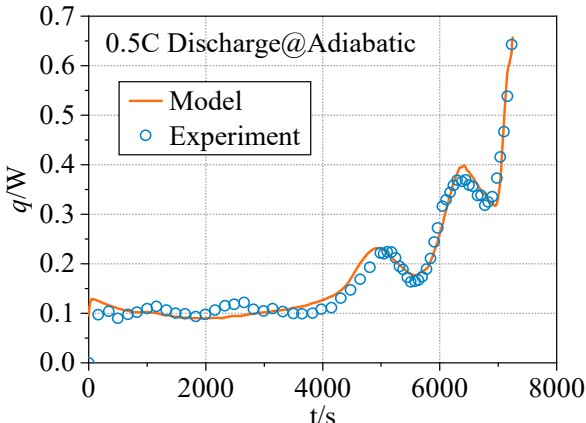

**Figure 5.** Validation of new battery discharge heat generation power under adiabatic conditions.

Figure 6 depicts the calculated findings of battery capacity decline under various operating conditions and the testing results. At the end of the age, when the battery was cycled at 23 °C C1D1 and 23 °C C1D2 working conditions, there was a dramatic decline in capacity. Lithium precipitation from the negative electrode and an increase in the rate of electrode active material loss could both be the real causes of the abrupt decline in capacity [31].

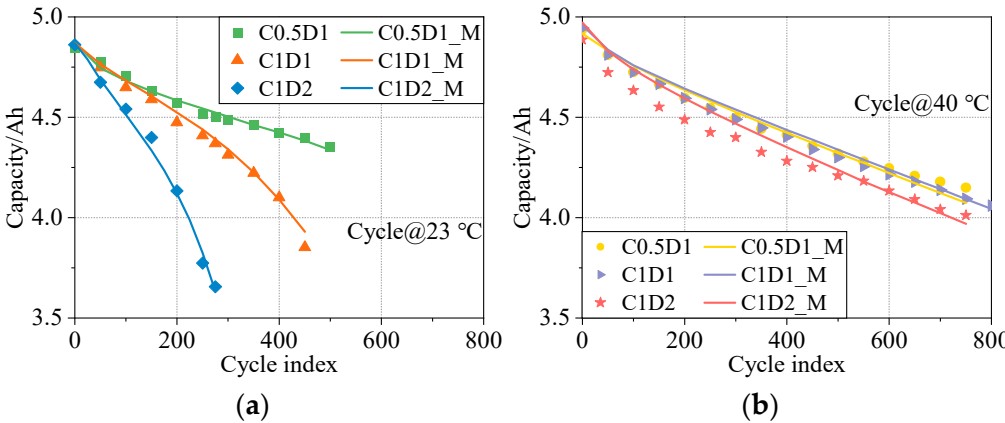

**Figure 6.** Validation of battery capacity under different conditions of cycle aging at (**a**) 23 °C and (**b**) 40 °C.

Since the heat exchange boundary conditions are consistent, the temperature rise during battery charging and discharging has a clear relationship with the battery's heat generation power, and the change in temperature could also be utilized to indicate the change in heat generation power. The experimental and computed results of the temperature rise on the battery's surface after aging were compared to validate the accuracy of the model developed in this chapter regarding the heat-generating characteristics of the battery after aging. Figure 7 depicts the temperature rise curves of the aged battery at different ambient temperatures during discharge under different cycling conditions. By comparing Figure 7a,b, it was found that the aged battery was more influenced by the ambient temperature at a low multiplier discharge, and the internal resistance of the battery decreased under a high-temperature environment (40 °C), resulting in lower temperature rises on the surface of the battery. Meanwhile, the simulation results of the discharge temperature rise of the aged battery at different temperatures were consistent with the trend in the experimental data, indicating that the model could be used for subsequent research.

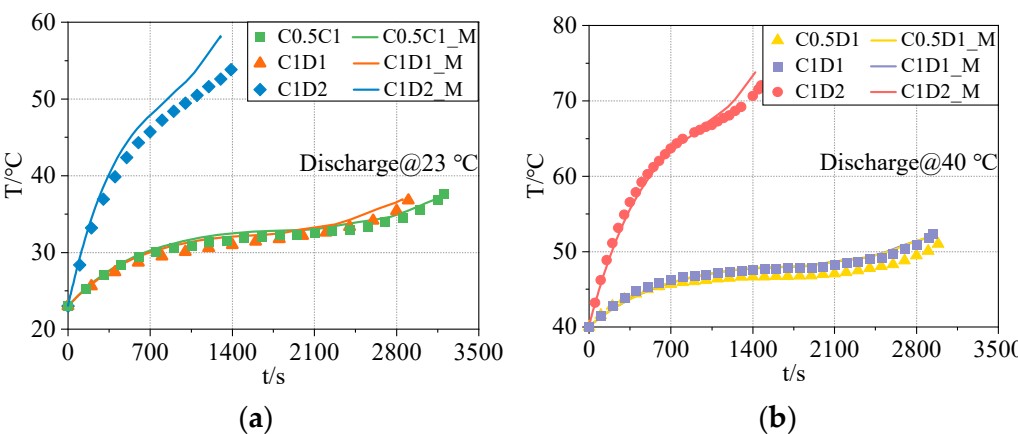

**Figure 7.** Validation of surface temperature rise of aging cells during discharge at (**a**) 23 °C and (**b**) 40 °C.

## 5. Results and Discussion

The performance and lifetime of lithium-ion batteries are particularly sensitive to temperature [34,35], and the majority of studies solely consider the accuracy of the model's computation of capacity loss, ignoring the changes in the heat-generating characteristics during battery aging. However, the different values of some key parameters are the cause for simulation calculation results that outline the changes in the heat generation characteristics of the battery aging process that deviate significantly from the actual situation; therefore, it is necessary to use simulations to study the influence of several key factors on the changes in heat generation in the battery aging process.

### 5.1. Changes in Heat Generation Power of Different Components during the Aging Process

Figure 8 depicts the heat generation power variation in different components with the number of cycles for the discharging process when the battery is cycled under 23 °C C1D1 and 40 °C C1D1 operating conditions. It can be observed that, as the battery ages, SEI heat generation increases the most, followed by negative solution phase ohmic heat, negative polarization heat, and positive polarization heat.

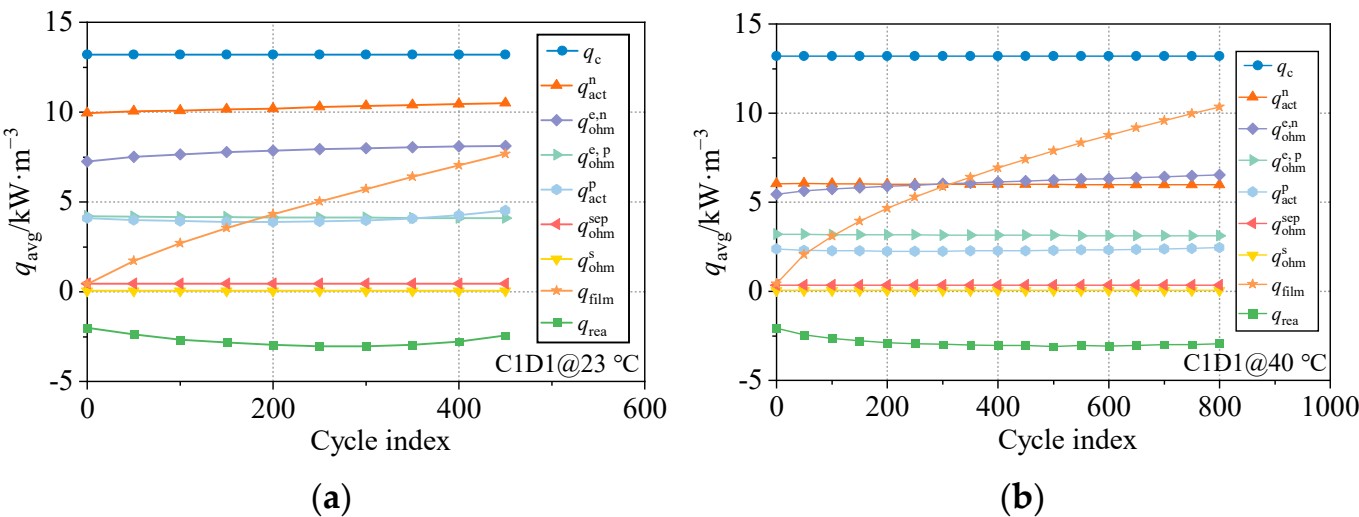

**Figure 8.** Variation in heat generation power for different components during cyclic aging at (**a**) 23 °C and (**b**) 40 °C.

As illustrated in Figure 9, the explanation for the rise in heat power in the SEI film is that the thickening of the SEI makes the path of lithium-ion diffusion from it longer, i.e., the resistance of the SEI increases. The reason for the increase in the ohmic thermal

power of the negative solution phase is that, on the one hand, the thickening of the SEI consumes the electrolyte, and on the other, its generation occupies the solution phase space of the negative electrode, resulting in a decrease in negative porosity [31]. In addition to the increase in SEI heat generation and negative solution phase ohmic heat power, the positive irreversible polarization heat at the end of aging increases slightly, which is related to the change in the window of electrode de-embedded lithium as a result of lithium-ion inventory loss and active material loss.

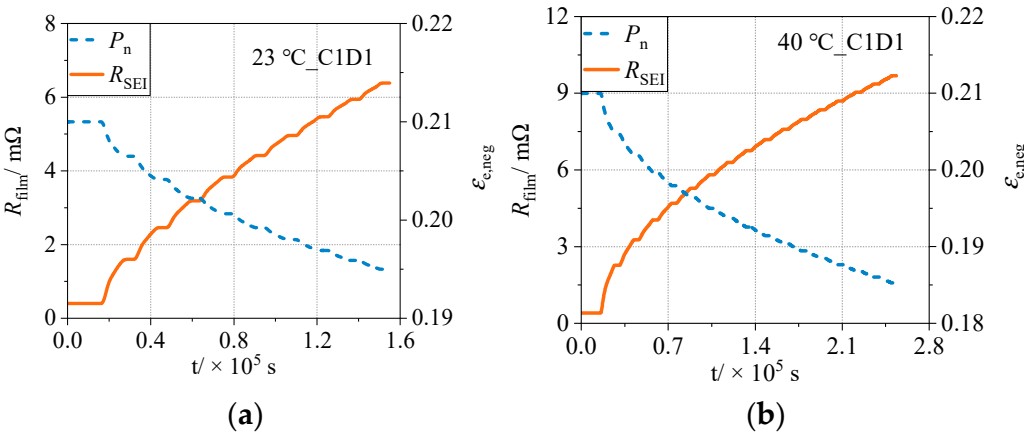

(a)

(b)

**Figure 9.** Variation in SEI resistance and negative porosity during cyclic aging at (**a**) 23 °C and (**b**) 40 °C.

### 5.2. Effect of SEI Physical Parameters on Heat Generation during Aging

The investigation of changes in the heat generation power of different components during the aging process revealed that the heat generation of the SEI growth reaction was the primary cause for the rise in heat generation power during the aging process of the cell. On the one hand, the ionic conductivity of the SEI directly determined SEI resistance, while on the other hand, its molar volume determined the growth of its own volume, which was caused by the consumption of a unit amount of substance by the lithium ions, and the volume growth of the SEI resulted in a decrease in the electrode's porosity, which impacted the negative electrode's kinetic performance [23].

#### 5.2.1. Ion Conductivity

Figure 10 depicts the battery capacity reduction, which was calculated for various SEI conductivity values [26,36–41]. It can be observed that the rate of capacity decline lowered as the ion conductivity of the SEI rose.

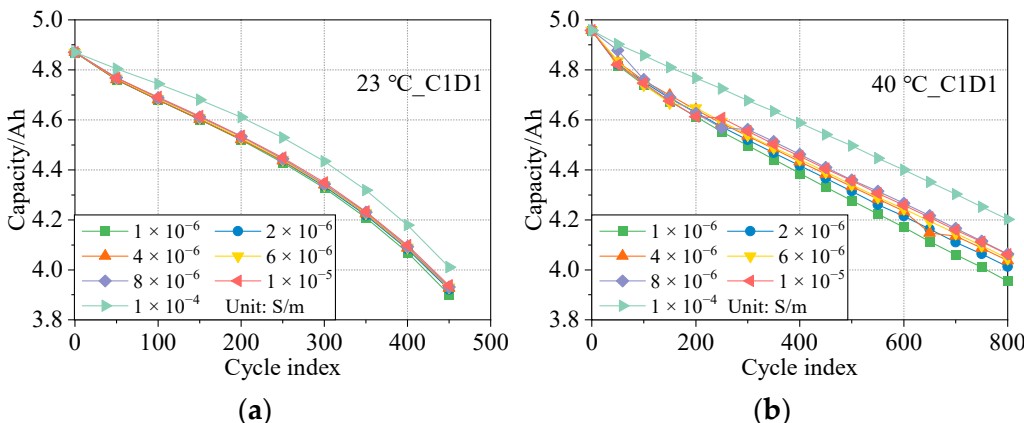

(a)

(b)

**Figure 10.** Simulation results of capacity decline when using different SEI ionic conductivities at (**a**) 23 °C and (**b**) 40 °C.

Figure 11 shows the SEI resistance and average negative porosity of aged cells for various SEI conductivity values. The results in the figure were derived from the discharge process' latest cycle. As the lithium-ion conductivity of SEI diminished, the resistance in the aged SEI membrane increased, particularly when it decreased from $1 \times 10^{-5}$ S/m to $1 \times 10^{-6}$ S/m, and the SEI resistance tended to accelerate growth. In addition, when the ionic conductivity of the SEI was reduced with age, negative porosity decreased.

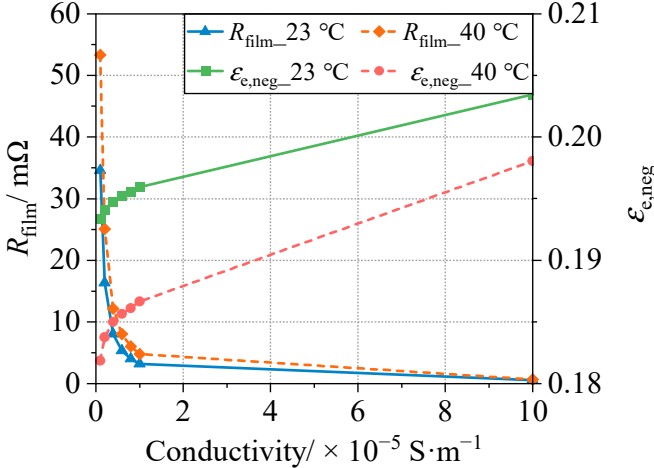

**Figure 11.** Simulation results of the SEI resistance and negative porosity of the cell after aging when using different SEI ionic conductivities.

As demonstrated in Figure 12, the substantial increase in SEI resistance led to a drastic increase in SEI heat generation power following cell aging. Additionally, when the ionic conductivity of the SEI membrane increased from $1 \times 10^{-5}$ S/m to $1 \times 10^{-4}$ S/m, there was no substantial increase in the SEI resistance and SEI heat generation at the conclusion of the cycle, showing that $1 \times 10^{-5}$ S/m was already a higher value for the SEI ionic conductivity.

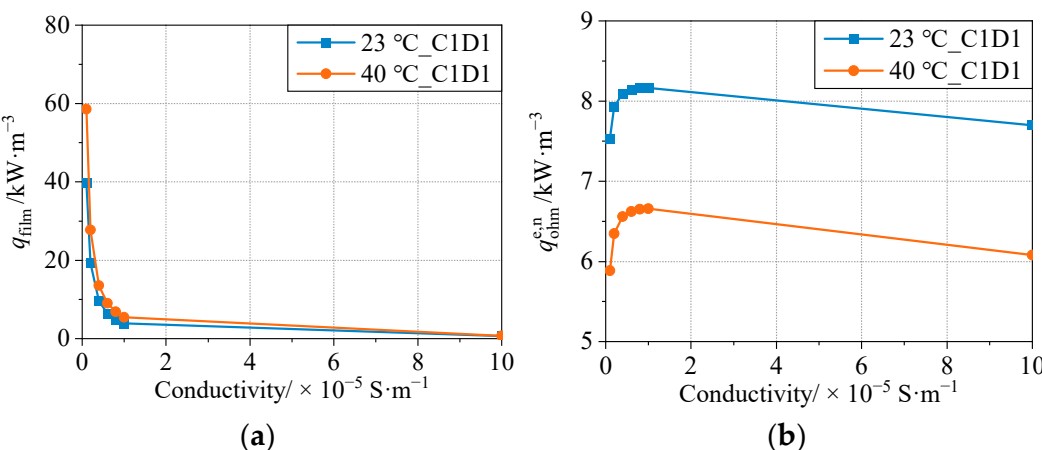

**Figure 12.** Simulation results of the heat generation rate during the discharge process of the cell after aging when using different SEI ionic conductivities. (**a**) SEI growth heat and (**b**) negative electrolyte ohmic heat.

Figure 13 demonstrates that as the ionic conductivity of the SEI decreases, the rate of increase in the heat generation power of the SEI during aging accelerates, resulting in a large increase in the maximum surface temperature rise during the discharge.

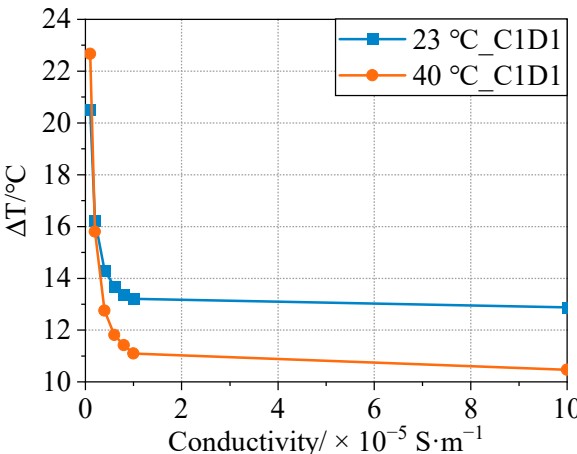

**Figure 13.** Simulation results of maximum surface temperature rise during the discharging process of the cell after aging when using different SEI ionic conductivities.

The preceding calculation results highlight the important value of SEI ion conductivity when conducting the coupled electrochemical–thermal–aging modeling of Li-ion batteries. Additionally, when the conductivity was between $1 \times 10^{-6}$ to $1 \times 10^{-5}$ S/m, the susceptibility to SEI resistance and cell heat production increased.

### 5.2.2. Molar Volume

Figure 14 depicts the battery capacity reduction calculated for different SEI molar volume values [25,28,36,37,39–41]. When the SEI molar volume increased, so did the rate at which the cell's capacity declined. Owing to the rise in the SEI molar volume, the SEI coating on the surface of the negatively active particles became thicker. The diffusion of solvent molecules in the electrolyte became challenging in the SEI, resulting in a decrease in the pace at which the SEI experienced a growth reaction; hence, the rate of cell capacity decline was lowered [42].

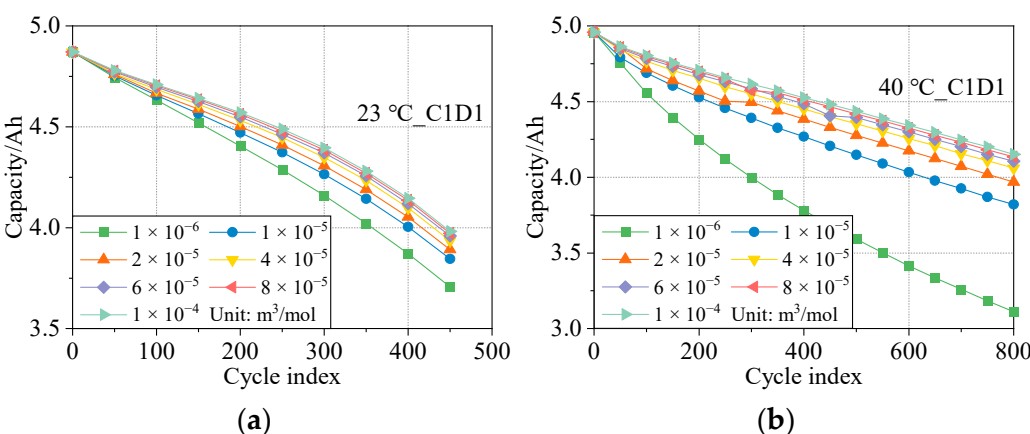

**Figure 14.** Simulation results of capacity loss when using different SEI molar volumes at (**a**) 23 °C and (**b**) 40 °C.

As illustrated in Figure 15, the rise in the molar volume of the SEI led to an increase in the SEI resistance and a decrease in the cell's negative porosity as a result of aging. An increase in the SEI resistance enhanced the SEI heat generation power at the same current. Figure 16 demonstrates that as the negative porosity decreased, the ionic conductivity and lithium-ion diffusion coefficient decreased while the resistance and ohmic heat of the negative electrolyte increased. As depicted in Figure 17, large increases in the heat generation of the SEI and negative electrolyte ohmic heat led to an increase in the temperature rise of the battery during operation after age. In contrast to the SEI's ionic conductivity, the SEI molar

volume values in the range of $1 \times 10^{-6}$ to $1 \times 10^{-4}$ m$^3$/mol had a substantial effect on the cell capacity's decline and heat generation after aging.

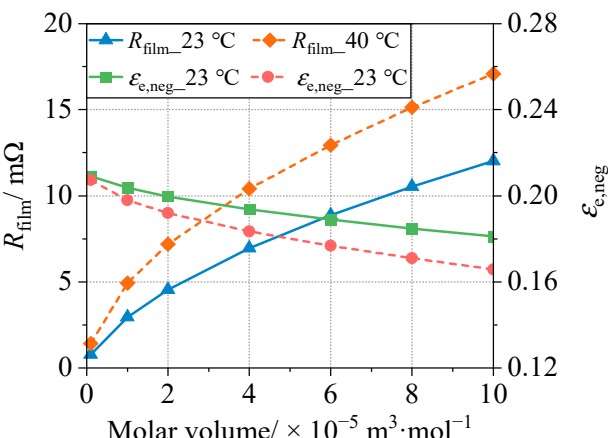

**Figure 15.** Simulation results of SEI film resistance and anode porosity of cells after aging when using different SEI film molar volumes.

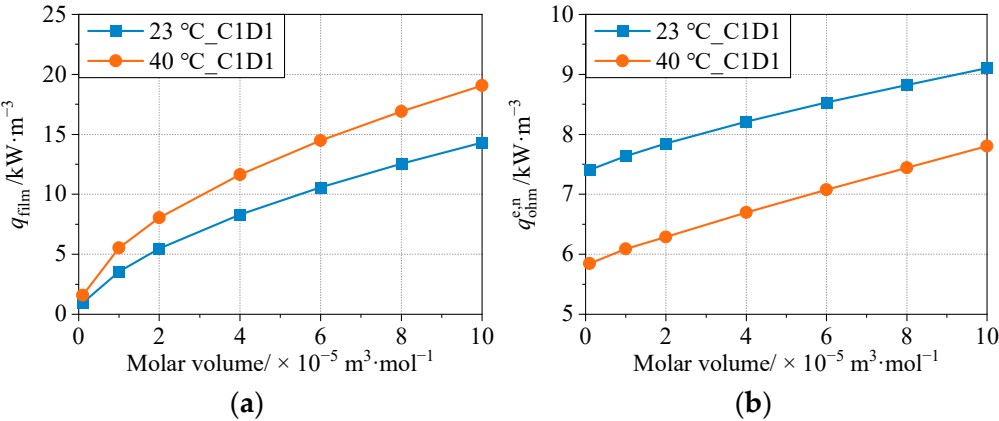

**Figure 16.** Simulation results of SEI and negative electrolyte ohmic heat generation rates during the discharge process of cells after aging when using different SEI film molar volumes. (**a**) SEI growth heat and (**b**) negative electrolyte ohmic heat.

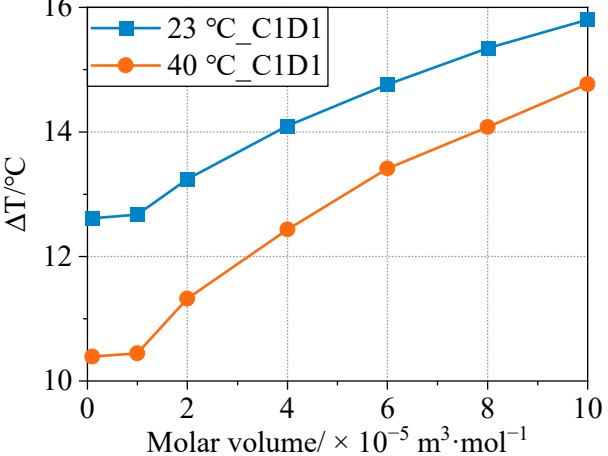

**Figure 17.** The maximum surface temperature rises during the discharging process of the cell after aging when using different SEI film molar volumes.

### 5.3. Effect of Aging Mode on the Change in Heat Generation during Battery Aging

The SEI heat generation, negative electrolyte ohmic heat, negative polarization heat, and positive polarization heat are the primary heat-generating factors that are altered with cell aging [9]; thus, the discussion focuses mostly on these four heat-generating terms. Figure 18 compares the average heat generation power of the primary heat generation terms at the beginning-of-life (BOL) and end-of-life (EOL) throughout the discharge process. The aging mode of an aging battery consists of only the LLI and concurrent LLI and LAM.

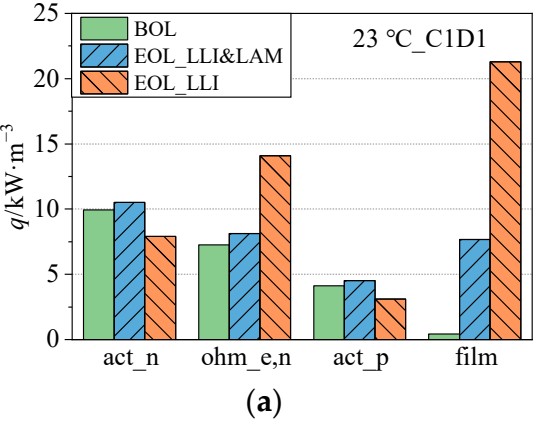 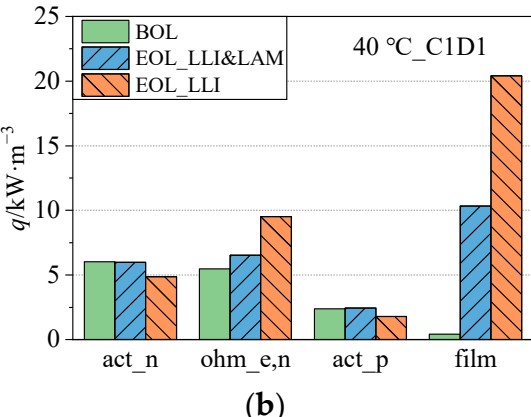

**Figure 18.** Comparison of the main heat generation items during the discharging process when the cell ages to EOL under different aging modes at (**a**) 23 °C and (**b**) 40 °C.

Figure 19 demonstrates that when just the LLI occurs in a battery, the negative electrolyte ohmic heat and SEI heat are greater after aging because all the lost lithium ions are involved in the SEI growth reaction, the SEI is thicker, and the porosity declines more after aging.

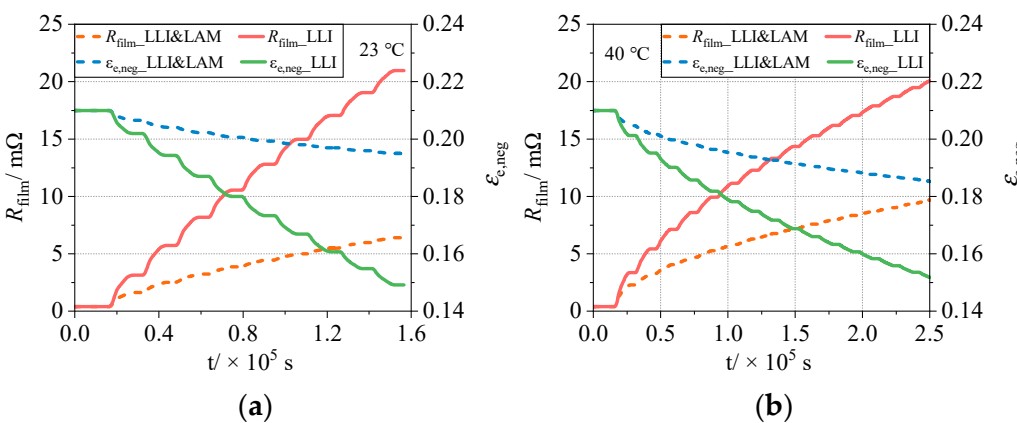

**Figure 19.** Variations in SEI film resistance and anode porosity under different aging modes at (**a**) 23 °C and (**b**) 40 °C.

In addition, only in the case of LLI was there a significant decrease in the heat of the polarization of positive and negative electrodes after aging in the battery. As shown in Figure 20, this is because the space available for lithium embedding in positive and negative electrodes remains unchanged, but the amount of lithium ions migrating by diffusion between positive and negative electrodes decreases, causing the maximum degree of lithium embedding in positive and negative electrodes to decrease.

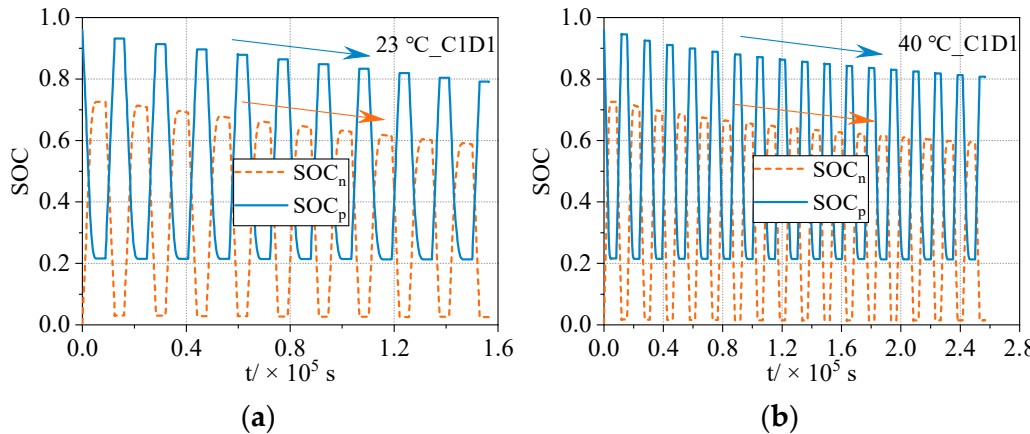

**Figure 20.** Variations in electrode SOC during the aging process when only LLI occurred in the cell at (**a**) 23 °C and (**b**) 40 °C.

In conclusion, if the real aging pattern of the battery contains LAM but is not accounted for in the modeling, it results in a big computation of SEI heat generation, negative electrolyte ohmic heat, and a minor calculation of polarization heat after the battery has aged. This is due to the greater sensitivity of SEI heat generation and negative electrolyte ohmic heat to the aging pattern, ignoring LAM results in a considerable increase in heat generation power and surface temperature after the cell has aged, as depicted in Figure 21. It can also be seen from Figure 21 that the contour of the temperature, represented by the increasing curve, during the discharge of the battery in different aging modes was distinct. When LLI and LAM occurred, the temperature rise rate at the conclusion of discharge was larger than when only LLI occurred. When the LAM is not taken into account, the maximum amount of lithium embedded in the positive and negative electrodes during the battery cycle drops, as does the polarization heat.

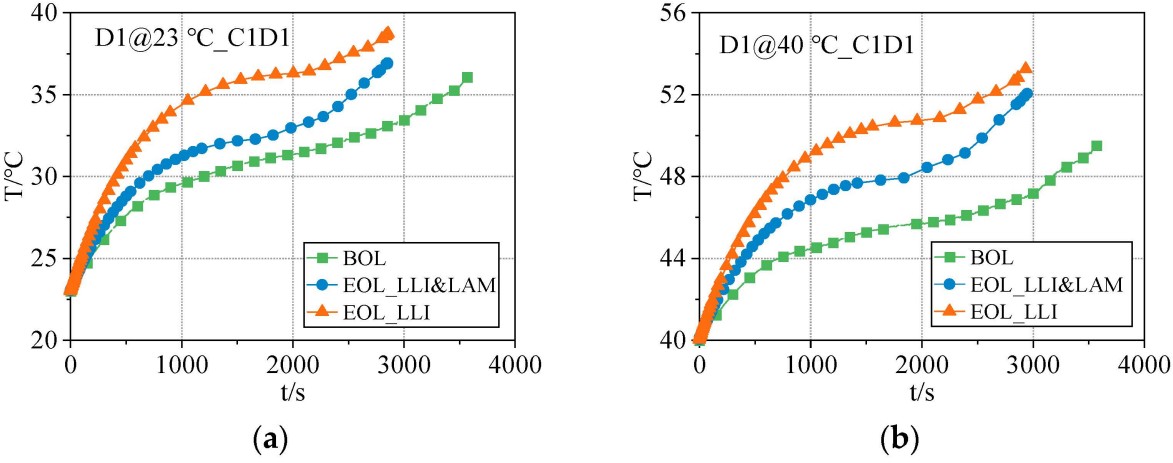

**Figure 21.** The comparison of the surface temperature rises during the discharging process when the cell ages to EOL under different aging modes at (**a**) 23 °C and (**b**) 40 °C.

## 6. Conclusions

In this study, based on the aging mechanism of LLI and LAM, a mathematical model of the electrochemical–aging–thermal interaction was constructed. The heat generation power and surface temperature rise of the battery were determined by comparing the simulation findings with the experimental data of combined working conditions with varying current charging and discharging rates (0.5 C, 1 C, and 2 C) and temperatures (23 °C and 40 °C). In terms of electrochemical characteristics, thermal properties, and aging properties, the simulation findings were validated and shown to be in good agreement with the actual

data. Based on this model, the heat generation characteristics of NCA batteries during aging were examined and the results revealed that:

(1) As the battery ages, its overall heat generation power at the same operating current grows dramatically. This is mostly due to the heat generation power increase in the SEI and the ohmic heat of the negative electrolyte.

(2) The SEI ionic conductivity falls below $1 \times 10^{-5}$ S/m, the SEI resistance and heat generation power grow exponentially, and the negative electrolyte ohmic heat diminishes quickly. The increase in the SEI molar volume causes both the SEI heat generation power and the negative electrolyte ohmic heat to rise. Therefore, in the battery manufacturing process, reasonable formation technology should be adopted to improve the ion conductivity of the SEI and accelerate the lithium-ion mass transfer on the electrode surface and, at the same time, improve the compactness of the SEI to inhibit the decrease in electrode porosity caused by volume growth.

(3) Compared with the occurrence of both LLI and LAM, when only LLI occurred in the cell, the SEI film was thicker, and the negative porosity lower when the same capacity declined, so the SEI heat generation and the negative electrolyte ohmic heat could be greater. Simultaneously, the maximum amount of lithium embedded in the positive and negative electrodes continued to diminish throughout the course of the cycle, as did the polarized thermal power at the conclusion of battery charging and discharging. Consequently, when just LLI occurred, the maximum temperature rise during the operation at the same capacity decline was greater than when both LLI and LAM occurred, and the temperature rise rate at the end of discharge was lower due to the narrower window of positive and negative lithium de-embedding.

**Author Contributions:** Conceptualization, R.H., Y.X. and F.C.; methodology, R.H., Y.X. and Q.W.; validation, Y.X., Q.W. and F.C.; formal analysis, J.C.; investigation, R.H., J.C. and X.Y.; writing—original draft preparation, R.H. and Y.X.; writing—review and editing, R.H., Y.X. and J.C.; visualization, Y.X. and Q.W.; supervision, R.H. and X.Y. All authors have read and agreed to the published version of the manuscript.

**Funding:** This research was funded by the State Key Laboratory of Clean Energy Use Open Fund, grant number ZJUCEU2022016.

**Data Availability Statement:** Not applicable.

**Conflicts of Interest:** The authors declare no conflict of interest.

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
