# Peer review of "Simulation Study on Heat Generation Characteristics of Lithium-Ion Battery Aging Process"

_electronics, doi:10.3390/electronics12061444_

Round 1
Reviewer 1 Report
This article described “Simulation study on heat generation characteristics of lithium- 2 ion battery aging process” the author is addressing the following questions; it would be helpful evaluating for further.
1. In fig 7. the author should explain how the temperature readings are measured, at which temperature increases rapidly, if it is increasing rapidly, what are the reasons for that, and clarify in detail. This is helpful to understand heat generation studies of Li-ion cells.
2. The data as you mentioned battery cell is missing, the author should include it and rewrite it again.
3. The cell capacity is not mentioned in the datasheet. Why is not mentioned, explain in detail and clarify a proper way.
4. The whole manuscript recommends polishing the language.

Author Response
Dear Reviewer,
I’m sincerely grateful to the insightful and constructive comments. They are very beneficial to improving the quality and readability of our paper. In the following, we carefully present our response to all the comments and highlighted the corresponding changes to our manuscript by using bold or colored text.
1. In fig 7. the author should explain how the temperature readings are measured, at which temperature increases rapidly, if it is increasing rapidly, what are the reasons for that, and clarify in detail. This is helpful to understand heat generation studies of Li-ion cells.
Reply: Thanks for the important questions. The temperature measurement method of fig 7. has been supplemented in this paper in “2. Experiments”. The temperature rise curve changes and the influencing factors of the battery discharging process in different ambient temperatures after aging under different cycling conditions in fig 7. have been introduced detailed in the corresponding position.
The manuscript is modified as follows:
A complete set of temperature measurement equipment consisted of a T-type thermocouple, NI 9214 thermocouple board, NI cRIO 9037 data acquisition controller, etc. The T-type thermocouple probe was stuck on the center site of the batteries,and the batteries were continuously charged and discharged with a certain constant current. Lastly, the surface temperature change during the operation was monitored. During the full process, the battery surface temperature was measured and recorded in situ by a multi-channel temperature logger.
Figure 7 depicts the temperature rise curves of the aged battery in different ambient temperatures during discharge under different cycling conditions. By comparing fig 7. (a) and fig 7. (b), it was found that the aged battery was more influenced by the ambient temperature at low multiplier discharge, and the internal resistance of the battery decreased under a high-temperature environment (40°C), resulting in lower temperature rises on the surface of the battery. Meanwhile, the simulation results of the discharge temperature rise of the aged battery at different temperatures were consistent with the trend in the experimental data, indicating that the model could be used for subsequent research work.
2. The data as you mentioned battery cell is missing, the author should include it and rewrite it again.
Reply: Thank you for your suggestion. The parameters used to build the model and the experimental data used for model validation in this paper are based on the study object 21700 NCA lithium-ion battery. To increase the readability of the paper, “2. Experiments” has been added to provide a detailed description of the study object.
The manuscript is modified as follows:
Commercial 21700 NCA battery (Φ21 mm × 70 mm cylinder battery, capacity 4.9Ah, Si-C anode material, Li (Ni0.8Co0.15Al0.05) O2 cathode material) is experimentally characterized.
3. The cell capacity is not mentioned in the datasheet. Why is not mentioned, explain in detail and clarify a proper way.
Reply: Battery capacity is an important data in battery-related research, and not stating the battery capacity is a shortcoming in the writing of the paper, which has been added in “2. Experiments” for the battery capacity of the 21700 NCA battery. The datasheet shows the parameters used in the electrochemical-aging-thermal modeling process. In the electrochemical modeling process, more attention is paid to the physical parameters of the materials inside the battery, such as the solid-phase lithium-ion concentration and the liquid-phase lithium-ion diffusion coefficient, etc. The establishment of the electrochemical model helps to understand the material distribution and change mechanism inside the lithium-ion battery from the microscopic level. The changes in battery capacity during charging and discharging can be obtained by combining electrochemical models and operating conditions.
4. The whole manuscript recommends polishing the language.
Reply: Thanks a lot for the reviewer’s comments. The manuscript language has been polished through MDPI's language editing services.

Reviewer 2 Report
REVIEWER COMMENTS
Manuscript ID: electronics-2286351
Simulation study on heat generation characteristics of lithium-ion battery aging process
General Comments:
The manuscript highlights the thermal management systems as the characteristics of heat generation from lithium-ion batteries as they age. Using the mathematical model, the authors studied various characteristics of 21700 (NCA) cylindrical lithium-ion batteries during ageing. The results emphasise at the same operating current, battery ageing greatly improves the cell's capacity to generate heat. The authors also discussed the formation of the solid electrolyte interface and its effect on the heat generation. The electrochemical principles underlying the effects of ageing on battery heat generation are comprehensively understood by looking at the changes in battery heat generation in various ageing phases. The simulation results are interesting that it is proven to be in good agreement in terms of electrochemical characteristics, thermal properties, and ageing parameters.
The authors have done a good attempt in evaluating the heat generation characteristics of Li-ion battery’s aging process which will give the experimentalist a different approach towards the problem.
Specific Comments:
· The authors are requested to give little more importance on the real time issue of solid electrolyte interface, its drawbacks in the first part of the introduction part
· How the authors validate the importance of molar volume in SEI heat generation?
· In the results and discussion part, the authors are requested to include reference to validate the obtained results
· Highlight how the present study helps to help the issue of SEI formation in the conclusion section
-----------------------------------------------------------------------------------------------------
Author Response
Dear Reviewer,
I’m sincerely grateful to the insightful and constructive comments. They are very beneficial to improving the quality and readability of our paper. In the following, we carefully present our response to all the comments and highlighted the corresponding changes to our manuscript by using bold or colored text.
1. The authors are requested to give little more importance on the real time issue of solid electrolyte interface, its drawbacks in the first part of the introduction part.
Reply: Thanks for the important suggestion. We have made corresponding changes in the first part of the introduction part.
The manuscript is modified as follows:
As a major side reaction leading to cell aging, the SEI growth reaction has an important influence on the heat generation of the cell during aging by its products [14-16].
Tang et al. [18] established an electrochemical-thermal coupling model for pouch cells, considering the electrolyte reduction decomposition reaction and the growth of SEI, and analyzed the changes in reversible heat, polarization heat, ohmic heat, and the total heat generation during the charging and discharging of batteries in different aging states; they found that the heat generation power increased during the charging and discharging of aged batteries, but that the total heat generation decreased.
2. How the authors validate the importance of molar volume in SEI heat generation?
Reply: Thank you for your question. The most intuitive manifestation in the SEI growth process is the volume growth, and the composition of the SEI product is very complex, including an inorganic layer composed of inorganic substances such as Li2CO3, Li-F, Li2O, and Li-Cl, and an organic layer composed of organic substances such as (CH2OCO2Li)2 and (CH2CH2OCO2Li)2. The SEI molar volume determines the growth of its own volume caused by the consumption of a unit material amount of lithium ions, and the growth of SEI volume leads to a decrease in electrode porosity, which affects the negative electrode kinetic performance. Therefore, molar volume was chosen as a variable to analyze the effect of SEI physical parameters on heat generation during aging.
The basis for the selection of SEI physical parameters has been added in the paper as follows:
The investigation of changes in the heat generation power of different components during the aging process revealed that the heat generation of the SEI growth reaction was the primary cause for the rise in heat generation power during the aging process of the cell. On the one hand, the ionic conductivity of the SEI directly determined SEI resistance, while on the other hand, its molar volume determined the growth of its own volume, which was caused by the consumption of a unit amount of substance by the lithium ions, and the volume growth of the SEI resulted in a decrease in the electrode porosity, which impacted the negative electrode kinetic performance [23].
3. In the results and discussion part, the authors are requested to include reference to validate the obtained results.
Reply: Thanks a lot for the reviewer’s comments. Relevant references have been added to the results and discussion part of the paper to increase the credibility of the results obtained.
Relevant references are as follows:
35. Z. Lu, X.L. Yu, L.C. Wei, F. Cao, L.Y. Zhang, X.Z. Meng, L.W. Jin, A comprehensive experimental study on tempera-ture-dependent performance of lithium-ion battery, Applied Thermal Engineering. 158 (2019) 113800.
36. W. Diao, S. Saxena, M. Pecht, Accelerated cycle life testing and capacity degradation modeling of LiCoO2-graphite cells, Journal of Power Sources. 435 (2019) 226830.
32. X.G. Yang, Y. Leng, G. Zhang, S. Ge, C.Y. Wang, Modeling of lithium plating induced aging of lithium-ion batteries: Transition from linear to nonlinear aging, Journal of Power Sources. 360 (2017) 28–40.
23. J. Christensen, J. Newman, A Mathematical Model for the Lithium-Ion Negative Electrode Solid Electrolyte Interphase, J. Electrochem. Soc. 151 (2004) A1977.
43. H. Ekström, G. Lindbergh, A Model for Predicting Capacity Fade due to SEI Formation in a Commercial Graph-ite/LiFePO4 Cell, J. Electrochem. Soc. 162 (2015) A1003.
4. Highlight how the present study helps to help the issue of SEI formation in the conclusion section.
Reply: Many thanks for this suggestion. We have perfected the conclusions in the manuscript to illustrate how the present study helps to help the issue of SEI formation.
The manuscript is modified as follows:
Therefore, in the battery manufacturing process, reasonable formation technology should be adopted to improve the ion conductivity of the SEI and accelerate the lithium-ion mass transfer on the electrode surface and, at the same time, improve the compactness of the SEI to inhibit the decrease in electrode porosity caused by volume growth.
Reviewer 3 Report
The authors of the paper have developed a numerical method to calculate continuously the impact of the aging process of the cylindrical lithium-ion batteries on the heat generation by taking into account also the solid electrolyte interface (SEI). They have shown that at the same electronic current, the heat generation power of the unit cell battery increases significantly mostly due to the battery aging process, among other important electrochemical or heat transfer mechanisms that cause the aging process. The important advantage of the proposed method is the use of the numerical model that accounts continuously for the power generation process as a function of battery life duration. This permitted the authors to study in detail the heat generation of batteries at different key parameters which are governed by the physical properties of the Solid Electrolyte Interface, inside the battery construction. By monitoring the battery aging process they have found the dependence of the battery voltage discharging and the battery surface temperature rise on the aging time. They obtained the optimal discharge temperatures which correspond to the C1D1, C0.5C1, C0.5D1 cycles agings.
The results in the paper are important and worth to be published. I strongly recommend the publication of the manuscript in the journal of Electronics.
Author Response
Dear reviewer,
thank you very much for reading this paper carefully and for your positive comments.